# Immunomodulatory Activity of the Tyrosine Kinase Inhibitor Dasatinib to Elicit NK Cytotoxicity against Cancer, HIV Infection and Aging

**DOI:** 10.3390/pharmaceutics15030917

**Published:** 2023-03-11

**Authors:** Andrea Rodríguez-Agustín, Víctor Casanova, Judith Grau-Expósito, Sonsoles Sánchez-Palomino, José Alcamí, Núria Climent

**Affiliations:** 1HIV Unit, Hospital Clínic-IDIBAPS, University of Barcelona, 08036 Barcelona, Spain; 2CIBER of Infectious Diseases (CIBERINFEC), 28029 Madrid, Spain; 3AIDS Immunopathogenesis Unit, Instituto de Salud Carlos III (ISCIII), 28029 Madrid, Spain

**Keywords:** dasatinib, tyrosine kinase inhibitors, CML, cancer, HIV-1, CMV, HIV functional cure, memory-like NK cells, γδ T cells, anti-aging, senolytic

## Abstract

Tyrosine kinase inhibitors (TKIs) have been extensively used as a treatment for chronic myeloid leukemia (CML). Dasatinib is a broad-spectrum TKI with off-target effects that give it an immunomodulatory capacity resulting in increased innate immune responses against cancerous cells and viral infected cells. Several studies reported that dasatinib expanded memory-like natural killer (NK) cells and γδ T cells that have been related with increased control of CML after treatment withdrawal. In the HIV infection setting, these innate cells are associated with virus control and protection, suggesting that dasatinib could have a potential role in improving both the CML and HIV outcomes. Moreover, dasatinib could also directly induce apoptosis of senescence cells, being a new potential senolytic drug. Here, we review in depth the current knowledge of virological and immunogenetic factors associated with the development of powerful cytotoxic responses associated with this drug. Besides, we will discuss the potential therapeutic role against CML, HIV infection and aging.

## 1. Introduction

### 1.1. Dasatinib 

Dasatinib is a broad-spectrum tyrosine kinase inhibitor originally developed to treat chronic myeloid leukemia (CML) [1]. CML is a hematopoietic progenitor cell leukemia, in which overgrown myeloid cells accumulate in bone marrow and peripheral blood [2]. A translocation between chromosomes 9 and 22 results in an aberrant chromosome 22 (or Philadelphia (Ph) chromosome), generating the BCR-ABL oncogene. BCR-ABL constitutively activates tyrosine kinases (TK) that drives both Ph+ CML and Ph+ acute lymphoblastic leukemia (ALL) [3]. Treatment for CML was drastically improved with imatinib, one of the first TK Inhibitors (TKI) targeting BCR-ABL TK [4]. Furthermore, development of these small molecules led to the development of alternative inhibitors such as the second-generation drug dasatinib, which yield up to 20-300 times higher activity and has faster and deep molecular response (DMR) than imatinib [1].

Dasatinib has other off-target effects inhibiting other TK (Table 1) [5] with potential side effects such as hematological, pulmonary and gastrointestinal toxicity that could limit its clinical use [6]. Interestingly, off-target effects have been related to potential benefits such as an increased natural killer cell (NK)-mediated cytotoxic capacity against cancerous cells and viral infected cells [7] and an anti-aging capacity. In this review, we will discuss the potential immunomodulatory effects of dasatinib on NK cells and other innate cells and its therapeutic role against CML, HIV infection and aging (Figure 1). 

### 1.2. Natural Killer Cell Biology

Natural killer cells (NK cells), are cells from the innate immune system that show strong cytolytic function against stressed cells such as tumoral cells and virus-infected cells. NK activation state is determined by a balance of multiple activation and inhibition signals mediated by NK inhibitory and activating receptors that bind to NK ligands from other neighboring cells (Figure 2A). Most of these ligands are HLA class I molecules, such as HLA-A, B, C and HLA-E. Depending on the balance between these NK receptors (NKR) and their HLA ligands, NK cells will either be activated to kill the target cell or inhibited, allowing the target cell to survive. NK cells are an heterogeneous population harboring multiple subsets that differentially expressed NK receptors such as: killer cell immunoglobulin-like receptors (KIRs), natural killer group 2 such as NKG2A, NKG2C and NKG2D, and natural cytotoxicity receptors (NCRs), for example NKp30, NKp40 or NKp46 [8,9]. The main activating and inhibiting NK receptors and their respective target cell ligands are shown in Figure 2A. The NKG2C and NKG2D receptors are activating NK receptors that could detect abnormal or malignant cells by binding with HLA-E and the MIC A/B ligands, respectively. These interactions potentiate NK cell killing [10,11] (Figure 2A).

NK cells are activated by target cells that down regulate “self” HLA. This NK activation mechanism appears by a phenomenon known as education or licensing. Education requires the interaction between inhibitory NK cell receptors and their own HLA ligands [13,14]. Educated NK cells remain inactive by the interaction of NKRs inhibitor with autologous HLA-expressing neighboring cells. However, NK cells can also be activated by viral infected cells and abnormal or injured cells that usually have a lower HLA class I expression to evade cytotoxicity driven by conventional CD8+ T cells. NK cells can detect these anomalous target cells by the lack of inhibiting interaction (low o non-inhibitory HLA interactions). This phenomenon is called “missing self” detection and generally results in NK cell activation, degranulation and target cell death (Figure 2B). For NK cell activation, additional activating signal are necessary [15]. NK cells are activated when the ratio of inhibitory and activator NKR signaling favors activation [9] (Figure 2B). Many inhibitory KIRs from NK cells interact with HLA alleles. KIR and HLA proteins are both encoded by highly polymorphic alleles, enabling a wide diversity of receptor/ligand interactions. Education through inhibitory receptors sensitizes NK cells to detect “missing self” cells, while education through activating receptors inhibited NK cell cytotoxicity [16] (Figure 2B).

Despite the high level of KIR gene content variability, there are two main groups of KIR haplotypes, termed “A” and “B” (Figure 2C). The A haplotypes include inhibitory KIR genes coding for receptors such as KIR3DL1, which recognize HLA-A and HLA-B molecules that contain Bw4 epitopes. The B haplotypes include several activating KIR genes such as KIR3DS1 (Figure 2C). Then, the KIR genetic combination can be AA, AB or BB genotype. There are two possible established epitopes for HLA-B: Bw4 or Bw6. KIR3DL1 is highly polymorphic and, depending on the haplotype, NK cells can express: high, low, or null levels of KIR3DL1 [17]. In fact, KIR3DL1 and HLA-B genetic polymorphisms regulate NK cell function. Their stronger ligand affinity correlates with enhanced NK cell inhibition but also with increased education and, consequently, potentiates “missing self” NK capacity to detect and kill anomalous cells [18] (Figure 2B).

The inhibitory NKG2A receptor binds to HLA-E molecules. Signal peptides from the leader sequence of HLA-A, -B and -C proteins, which are codified by the first exon of the MHC gene, must bind to HLA-E in order to fold properly and reach the cell membrane and to become a ligand for the inhibitory NKG2A receptor. In addition, the anchor residue of the nonamer peptides that bind to HLA-E corresponds to residue -21 of the classical HLA class I leader sequence. There are two amino acid variations at position -21, a methionine (-21M) or a threonine (-21T) [19,20]. HLA-B antigens can have both -21M and -21T variants, depending on the HLA-B haplotypes that are associated with two NK cell education profiles. On the one hand, HLA-B -21M alleles contribute more effectively to NKG2A-mediated education, which suppresses NK cell activation when the target cell expresses stabilized levels of HLA-E. HLA-B -21T alleles mainly contribute to education through inhibitory KIRs, especially KIR3DL1 [15], as HLA-B -21T and HLA-Bw4 are genetically linked. Then, the HLA-Bw4 alleles are, in fact, HLA antigens encoded by -21T, able to interact and educate through KIR3DL1/HLA-Bw4, allowing the NK cell cytotoxicity upon “missing self” recognition [9,20] (Figure 2B).

## 2. Dasatinib-Mediated Immunomodulatory Effects in CML

### 2.1. Dasatinib Effect on NK Cells and Innate T Cells

Besides its direct impact on leukemic cells through inhibition of constitutive tyrosine kinase activity, dasatinib’s off-target effects include: inhibition of proliferation and activation of T cells, and in vitro suppression of cytotoxic activity of NK cells [21,22] This is due to the potent inhibition of several off-target kinases of the Src family, such as Lck and Fyn in T cells [23]. TKI treatments can restore an anti-leukemic effector function, such as specific cytotoxic T lymphocytes’ (CTL) responses against leukemia-associated antigen (LAA). Overall, this results in a reduction in leukemic cell load major molecular remission (MMR) and a better recovery [24,25]. Dasatinib has the potential to reduce Tregs and derived factors (sCTLA-4), especially in patients developing large granular lymphocytes (LGLs), lymphocytosis [26] and NK cell differentiation, promoting immune stimulation [27]. This is relevant because the proportion of Treg cells is abnormally elevated in CML individuals at diagnosis, compared to healthy controls. Furthermore, dasatinib also reduces myeloid-derived suppressor cells (MDSC) in CML patients [28].

Up to half of the patients receiving dasatinib treatment develop an LGL lymphocytosis, mainly composed of cytotoxic cells (NK cells and CD8^+^ γδ T cells) [29,30] but this also includes CD57^+^ cytotoxic CD4^+^ T cells with anti-leukemic properties [31]. This LGL expansion is associated with an improved anti-leukemic response in both CML and Ph+ ALL patients [29,32,33,34,35]. The CD8^+^ T cell response in dasatinib-mediated LGL response includes TCR-Vβ^+^ expansion of either oligoclonal or polyclonal origin with cells resembling healthy memory CD8^+^ T cells [36]. Importantly, dasatinib may have an immunomodulatory role in non-conventional or innate T cells by increasing cell number, activation status and Th-1 polarization of innate αβ iNKT-cells in treated CML patients [37]. In addition, it has been reported that dasatinib increased a novel innate subset termed innate CD8^+^ T-cell-expressing IFNγ [37]. 

CML patients treated with dasatinib presented more classical NK cells (CD3^−^CD56^+^) and matured NK cells (CD56^+^CD57^+^) compared to imatinib- or nilotinib-treated patients [38]. These patients also presented lower expression of NK-inhibitory markers (KIR2DL5A, KIR2DL5B and KIR2DL5), which was associated with an MMR [39] and an increase in KIR2DL1 expression [8].

Dasatinib promoted NK-cell cytokine expression and cytotoxic activity towards the CML-derived cell line K562 [18,40], especially when KIR3DL1/HLA-Bw4 interactions were null, low or weak. Of note, Izumi et al. showed that blocking KIR3DL1/HLA-Bw4 binding with an anti-KIL3DL1 antibody potentiated NK cytotoxicity [18]. Furthermore, Shen et al. reported that this increased NK cytotoxic activity can be emulated in vitro using dasatinib treatment after IL-2/IL-15-mediated expansion of NK cells. In this setting, dasatinib also increases the percentage of NKG2A^−^CD57^+^ NK cells and the expression of activating receptors CD226 (DNAM-1), NKp46 and NKG2D [41], promoting their capacity to kill by degranulation of the CML cell line K562, not expressing HLA class I. Finally, dasatinib-mediated reduction of NKG2A also may boost NK cytotoxicity and improve MMR [18,42].

### 2.2. Dasatinib Increases Memory-like Natural Killer (NK) Subsets Displaying Activity against Both Leukemic and Cytomegalovirus (CMV) Infected Cells

There is discussion on whether development of LGLs with dasatinib is associated with previous immunity to CMV [32,43] or related to CMV viral load (VL) [7,43]. Ishiyama et al. reported that NK cells are the main component of LGLs in patients with CML treated with dasatinib, and NK cell expansion was highly associated with CMV-serostatus [7]. The authors performed a principal component analysis (PCA) with multiple markers on NK cells after dasatinib treatment and determined that NK cells from CMV^+^ individuals had a CMV-associate phenotype, named memory-like NK cells, with highly differentiated NK phenotypes (NKG2C^high^NKG2A^low^CD57^high^LIR-1^high^NKp30^low^NKp46^low^). NK cells from CMV-uninfected individuals were negative for this CMV-related signature [7,44]. Remarkably, a higher grade of NK cell differentiation at CML diagnosis predicts both a greater expansion of CMV-related memory-like NK cells and a lower leukemic-cell load after dasatinib treatment in CMV^+^CML patients [7,44]. Kadowaki et al. reported that a persistent, low-level CMV replication, often subclinical, triggered memory-like NK cell expansion. This suggests that CMV may trigger NK-cell expansion and both leukemia and dasatinib are enhancing factors that expand this NK subpopulation [45]. This hypothesis was reviewed by Climent et al. [46]. Importantly, this NKG2C^+^CD57^+^ subset of memory-like NK cells may represent NK cells with unique adaptive and editing properties [44]. Thus, the expansion of this subset could promote long-term memory and high cytotoxic activity against CML, even after dasatinib treatment interruption [45]. Recent studies have also observed an expansion of CD56^neg^ NK cell populations exclusively in CMV^+^ patients treated with dasatinib, and this increase parallels memory NKG2C^+^CD57^+^ NK cells [47]. These authors propose that CD56^−^ NK cells may be an exhausted population induced by chronic activation through CMV reactivation but, paradoxically, are proposed as a hallmark of CML control because it predicts a better clinical outcome. Overall, these results suggest that dasatinib immunomodulatory effects on NK cell responses against CMV could also be relevant against malignancies or other viral infections [48], such as HIV infection [46].

### 2.3. CML Control and Therapeutic Treatment Interruption: Immunological Factors Involved in a Successful Treatment-Free Remission (TFR)

Once a DMR is observed, TKI treatment can be interrupted in selected patients with the aim of achieving a TFR [49]. Dasatinib and nilotinib treatments are associated with a stronger DMR, increasing the chances of longer TFR [49]. A better understanding of the factors that could predict longer TFR is of paramount importance [50,51,52]. Moreover, half of the individuals who stopped treatment were able to keep TFR during at least one year and developed high levels of NK cells [53,54,55] and neutrophils [56] but not T cells [53,57] indicating that these populations are key to keeping CML under control.

Regarding the implication of NK cells in TFR, individuals who controlled CML after imatinib therapy cessation showed higher NK cytotoxic function towards the target K562 cell line, lacking HLA class I. Furthermore, NKG2D gene polymorphisms [58] and the IFN-γ and TNF-α cytokine secretion by NK (CD56^dim^CD16^−^) cells correlated with the successful drug discontinuation and control of CML [57,59]. Increased mature (CD57^+^) and cytotoxic (CD16^+^ and CD57^+^) NK cells, together with IFN therapy prior to TKI cessation, have been also shown to produce better CML outcomes after treatment interruption [60]. IFNα treatment also increased differentiated NKG2C^+^ NK cells, increased NKp46 expression on the CD56^bright^/CD16^-^ NK cell subpopulation and modulated NK cell cytotoxicity [61].

TFR has also been evaluated in dasatinib stopping TKI trials [62]. The DADI study [53,63] demonstrated that high levels of NK cells (CD56^+^), LGL NK cells (CD56^+^CD57^+^) and low levels of Treg (CD25^+^CD127^low^) preceding the dasatinib interruption were associated with longer TFR periods. Furthermore, a critical role of Treg inhibition by dasatinib, potentiating NK cell function, promotes a DMR [64]. Likely expanded NK cell functionality and lower Treg frequencies may decrease the probability of a worse outcome after dasatinib interruption and result in longer TFR [65]. 

Recent studies suggest that the features most consistently linked to longer TFR, independently of the type of TKI treatment stopped, are: (1) a high frequency of cytotoxic subsets such as NK, NKT and CD8^+^ γδ T cells [66,67]; (2) high level of NK-activating receptors such as NKG2D, NKp30 or NKG2C on NK [11,67,68] and NKT cells [67]; (3) enhanced expression of activation cytokines or granzyme B in NK cells after stimulating with HSP70; and (4) KIR homozygosis at haplotype AA, which includes KIR3DL1. These hallmarks may be useful as prognostic biomarkers of longer TFR [67]. Further clinical trials are needed to test these predicting biomarkers for TFR [69]. Moreover, the memory-like NK cells, characterized as CD3^−^CD56^dim^CD57^+^ NKG2A^−^ NKG2C^+^, were increased in patients with TFR success [70]. 

Results concerning NKG2A expression on NK cells are controversial [11]. Several studies suggested that low levels of NKG2A expression in NK cells is associated with longer TFR [70] and better CML prognostic [42]. In stark contrast, Xu et al. reported that an elevated expression of NKG2A in NK cells, especially in the CD56^bright^ subset, was a good prognostic biomarker for TFR [11], as also reported by Vigón et al. [67]. The inhibitor receptor NKG2A has a dual function on NK cells: firstly, it has a key function in the NK education process; second, after inhibition of NK cell activation it could send other signals to NK cells [71]. In fact, we recently found that high expression of NKG2A is present in NK cells able to kill cancerous cells such as reprogrammed cells or those that downregulate HLA-E. Reprogrammed cells express Yamanaka factors and this gives them the capacity of being pluripotent embryonic stem cells (e.g., teratoma-like cells). These educated NK cells are able to kill by the missing self-recognition [72]. NKG2A expression could not be per se detrimental to the function of all NK cell subsets. Altogether, the increased expression of NKG2A in the CD56^bright^ subset could be interpreted as an increased killing capacity against target cells with lower expression of HLA-E, such as cancerous cells or CML cells [11].

### 2.4. NK Immunogenetics Associated with CML Control or TFR

The diversity of allotypes of KIR3DL1 and HLA-Bw4 is associated with the receptor/ligand avidity and the NK cytotoxic capacity [12,17,73,74]. KIR genotypes have high genomic variations or allotypes that are associated with NK cell cytotoxic activity against CML and extended TFR periods [75,76]. Some reports suggested that CML patients with KIR2DL5B and KIR2DL2 alleles reached higher DMR after TKIs, implying that some KIR alleles or a specific combination of KIR genes can modify NK cell activity against CML cells [77,78]. In fact, KIR AA haplotypes, which include many KIR inhibitors such as KIR3DL1, are associated with better outcomes in TKI-treated CML patients [77] and are also linked to patients with sustained TFR [8,75]. It is interesting that the A haplotype including KIR3DL1 is highly associated with NK education (Figure 2B,C). Interaction of KIR3DL1 and HLA-Bw4 could affect NK cell education and cytotoxicity. In fact, the KIR3DL1*005 allele was highly linked with DMR, suggesting the relevance of this specific KIR3DL1. Consistently, DMR was coupled with higher NK cell killing in vitro in a NK cell cytotoxic assay against the CML cell line K562 without HLA class I, indicating that these educated NK cells could contribute to eliminating CML cells in vivo (Figure 2B). However, verification is needed, and the specific A haplotype KIR gene with the greatest impact needs to be identified [12] (Figure 2B). It has been shown that haplotypes of KIRs and HLAs were linked to a better outcome in a Japanese cohort [18,76]; specifically, Ureshino et al. reported that TFR in patients with HLA-Bw4 was higher than in patients with HLA-Bw6 alleles [79]. Similarly, HLA-Bw4 has been related to HIV control [80]. Altogether these genetic associations suggest that NK cells from patients achieving DMR or TFR could be better educated by interaction between the inhibitor receptor KIR3DL1 and HLA-Bw4, allowing activation of NK cells against cancerous cells that downregulate HLA expression, triggering “missing-self activation” (Figure 2B). Clinical trials with large cohorts are needed in order to explore deeply the NK immunogenetic factors associated with CML control [12]. 

### 2.5. Dasatinib as an Immunomodulator in Other Therapeutic Strategies against Cancer

The immunomodulatory activity of dasatinib has also been evaluated in combination with other therapeutic strategies against advanced malignancies such (1) immunotherapy with immune checkpoint inhibitors, where dasatinib immunomodulatory capacity has been investigated in increased programmed cell death protein 1 (PD-1) and programmed cell death ligand 1 (PD-L1) (PD1-PDL1) immunotherapy [81,82]; or (2) with chimeric antigen receptor (CAR)-engineered T cells (CAR-T). In CAR-T therapies, T cells are ex vivo modified by adding a gene for a receptor that helps the T cells to target specific myeloid antigens. A combined CAR-T and TKI approach has also been evaluated in some studies to enhance antitumor immunity and demonstrated that dasatinib limits CAR-T cells’ therapy side effects, such as the cytokine release syndrome (CRS) [83], and increases the anti-leukemia activity of CAR-T cells by decreasing cell exhaustion [84]. 

### 2.6. Summary of the CML Section

In brief, these findings support the idea that dasatinib contributes to better treatment response in CML patients through enhancement of the immune system, particularly via NK cell differentiation. CML patients with better outcomes could have done better due to genetic factors, such as AA alleles (homozygosis at KIR3DL1) and HLA-Bw4 associated with educated and highly cytotoxic NK cells able to detect malignant cells. Consequently, dasatinib could be useful, especially in the patients that do not have these protective features, to enhance cytotoxic activity of NK cells against CML cells by increasing memory-like NKG2C^+^CD57^+^ NK cells [70], γδ T cells and other innate CD8^+^ T cells [7,30,37,44]. In addition, these innate cells could express high levels of the activation receptor NKG2C, NKG2D, NKp46 or DNAM-1 and downregulate some inhibitory receptors such as NKG2A and KIR2DL5 [7,39,41,44] (Figure 1A). 

## 3. Potential Use of Dasatinib in the Setting of HIV-1 Infection

Current antiretroviral therapy (ART) can prevent progression to AIDS, blocking new infections by interfering with the virus life cycle. ART efficiently decreases the plasma VL under the limit of detection of conventional techniques (50 copies/mL) but is unable to fully eliminate HIV from the body. Latently infected cells, established during the early stages of infection, harbor integrated forms of the virus that are responsible for viral rebound when ART is stopped. These viral reservoirs represent the major barrier to the complete eradication of HIV, because it is invisible to the immune system and inaccessible to treatment [85]. In recent years, several strategies have been proposed to eliminate these reservoirs, including the ‘shock and kill’ strategy, but none of them have demonstrated a significant decrease of the reservoir size [86]. Therefore, new strategies are needed to avoid the formation of the reservoir, but also its replenishment and maintenance, by additional mechanisms including immunotherapy and new immunomodulatory compounds [87].

### 3.1. Effect of Dasatinib on HIV Infection and Reservoir: Direct Effect

Dasatinib directly interferes with CD4^+^ T-cell activation, which is the main HIV-1 cell target, by inhibiting the Src TK implicated in T cell receptor (TCR) signaling and the phosphorylation of SAM domain and HD domain-containing protein 1 (SAMHD1), a cell restriction factor. Here, we discuss the potential use of dasatinib as an adjuvant of the ART during HIV-1 infection. This strategy might reduce the viral reservoir and control infection [87,88,89]. Quite unexpectedly, CD4^+^ T cells from CML individuals under dasatinib therapy were shown to be resistant to ex vivo HIV-1 infection [88,90]. Results from our own work showed that dasatinib preserved SAMHD1 activity, by maintaining its active conformation, which restricts HIV-1 replication [46,87,88,89]. This enzyme controls homeostatic balance of dNTPs. When active (unphosphorylated), SAMHD1 restricts HIV-1 replication by lowering dNTP to levels that do not allow an effective viral replication. In contrast, when SAMHD1 is inactive (phosphorylated), it increases intracellular dNTPs’ availability, facilitating HIV-1 replication [46,87,88,89]. Moreover, Williams et al. described how the inhibition of cyclin-dependent kinases (CDKs) 1, 2, 4 and 6 in macrophages by dasatinib leads to blockade of HIV-1 infection by dephosphorylation of SAMHD1. This confers protection from the virus not only in the CD4^+^ T cells but also in monocyte-derived macrophages [91]. 

Dasatinib targets multiple host non-receptor tyrosine kinases, in particular the ABL family of kinases such as ABL1 or ARG. It has been demonstrated that siRNA knockdown of ABL1 and ARG kinases inhibits HIV-1 at a step after reverse transcription during early infection of activated CD4^+^ T cells, increasing the unintegrated two-long terminal repeat DNA circles (2-LTRcs) generated through the ligation of the cDNA ends by the host cell non-homologous DNA end-joining system. These results suggest preliminary efficacy of dasatinib treatment during acute HIV-1 infection [92]. Meanwhile, Bermejo et al. and Vigón et al. found that the frequency of proviral integration was strongly reduced in activated CD4^+^ T cells from CML patients treated with dasatinib compared with those from untreated healthy donors [88,93]. Importantly, we collaborated to find that peripheral blood lymphocytes obtained from CML patients treated for at least 6 months with dasatinib and individuals that were in TFR for more than one year were resistant to HIV-1 infection [46,93]. This indicates that mechanisms other than SAMHD1 phosphorylation are implicated and suggests an immunomodulatory role of dasatinib. 

Despite CML and HIV-1 infection not being commonly associated, several cases of people living with HIV(PLWH) who developed CML have been described. Studies performed on this special subset of HIV^+^ individuals treated with ART and dasatinib showed that the frequency of latently infected cells was reduced more than 5-fold compared with PLWH only on ART. In addition, reactivation of proviruses in CD4^+^ T cells isolated from PLWH with CML on ART and dasatinib was reduced 7-fold compared with individuals only on ART. Importantly, dasatinib treatment dramatically impaired SAMHD1 phosphorylation *both in vitro and in vivo* (30-fold and 21-fold, respectively) [94]. Dasatinib also blocks CD4^+^ T cell proliferation induced by homeostatic cytokines such as IL-7 and IL-15, which are crucial for the stability of the viral reservoir. Innis et al. described the restriction of both homeostatic and antigen-driven proliferation in memory CD4^+^ T cells from PLWH by dasatinib, consistent with promoting a smaller reservoir size [95]. In addition, the antimitotic properties associated with dasatinib can reduce the clonal expansion of infected cells, limiting the permanent filling of viral reservoirs [87]. 

### 3.2. Dasatinib’s Indirect Effect in HIV-1 Infection: Potentiation of NK and γδ CD8^+^T Cell Responses

Vigón et al. suggested that potentiation of memory-like NK cells and a γδ CD8^+^ T cell responses in dasatinib-treated CML patients may contribute to the observed resistance of these individuals’ PBMCs to HIV-1 infection. This mechanism would operate in parallel to the previously described phosphorylation of SAMHD1 [93]. Interestingly, this increased cytotoxic activity of memory-like NK cells and a γδ CD8^+^ T cell subsets might be effective both against HIV infection and CML. A similar cytotoxic cell signature has recently been described in a case of exceptional HIV long-term post-treatment control (>15 years, Barcelona patient). This control has been linked to a strong memory-like NK cell response, with NKG2C^+^CD57^+^ phenotype and γδ cytotoxic CD8^+^ T cells [96]. This NK cell phenotype (NKG2C^+^CD57^+^) was also found in long-term HIV elite controllers [97] and in individuals during primary infection who showed lower HIV RNA expression and more quickly reached an undetectable VL [98,99,100]. The above phenotype has been described together with a reduction of KIR2DL5, NKG2A (inhibitory receptors) and NKp30 in NK cells [93,97]. 

Taking all this into account, it could be considered that, despite immune exhaustion in PLWH, dasatinib is able to increase NK memory-like (CD56^+^CD57^+^) and γδ CD8^+^ T cells subsets and to reduce NKG2A expression in NK cells [42], contributing to HIV-1 infection control.

### 3.3. NK Cells’ Role in HIV-Mediated Control and Functional Cure

Functional HIV cure, defined as control of viral replication in the absence of treatment, has been described in two scenarios: in those PLWH who maintain undetectable HIV VL (<50 copies/mL) in the absence of ART, known as “elite controllers” (EC), and those individuals who demonstrate sustained virologic suppression for months or years after treatment cessation, called “post-treatment controllers” (PTC). The definitions of these patients vary depending on the time elapsed without treatment, since in many of them a “loss of control” occurs over time, defined by an increase in VL, which makes it necessary to restart ART. The patients of greatest medical and biological interest are those who have controlled the infection for at least ten years in the absence of ART and will be called long-term or exceptional elite controllers (LTEC or EEC) and long-term post-treatment controllers (LTPTC). This rare subset of patients may serve as a realistic model of a functional cure for HIV-1. LTPTC have a NK-mediated HIV control [96]. Most LTPTC are patients treated in the acute infection phase in whom spontaneous control of the viral load is observed when the treatment is discontinued months or years later [101,102]. The percentage of LTPTC patients is difficult to estimate and although some authors put it at 10%, it is probably lower [101]. 

### 3.4. Immunogenetic Factors of HIV Control and Functional Cure

The most relevant genetic alleles that are associated with HIV control are the HLA-B allotypes, such as HLA-B*57 and HLA-B*27, both being members of the HLA-Bw4 epitope [103]. Interestingly, HLA-Bw4 homozygosis is associated with HIV control and protection [80,104,105,106]. HLA-A expression levels change from one allotype to another [107]. Decreased HLA-A expression correlates with HIV control, being seen in the EC patients that have reduced HLA-A levels [107]. In addition, -21T amino acid variation at the signal peptide HLA-B has been associated with being HLA-Bw4 [9,20]. For that reason, homozygosis at HLA-B -21T, likely associated with Bw4 homozygosis, contributes to HIV control [80]. Some KIR3DL1/Bw4 combinations also are associated with HIV control [9]. In this context, HLA alleles carrying HLA-B -21T, such as Bw4 and low-expression HLA-A allotypes, codify HLA molecules that reduce HLA-E levels -n HIV-infected CD4^+^ T cells. Once KIR3DL1/HLA-Bw4-educated NK cells interact with HIV-infected target cells expressing low levels of HLA-E, activation of NK cells by lack of inhibitory signaling could be able to enhance cytotoxic function against HIV-infected cells, thus promoting HIV control [9,107] (Figure 3).

Consistently, in vitro experiments have suggested that KIR3DL1/HLA-Bw4 genotype combinations have higher educated NK cytotoxic responses than NK cells from Bw6 homozygotes against HLA-downregulated target CML cell lines (K562) [9,108,109,110] or HIV-infected CD4^+^ T cells that decreased the levels of HLA-A, -B and -C by Nef and Vpu HIV proteins [111,112,113,114] emulating the missing-self NK cytotoxic activity (Figure 3) [4,114,115]. Moreover, a higher number of educated NK cells by KIR2DL1/HLA-C2 and KIR2DL3/HLA-C1 interactions can mediate the killing of K562 cell lines and HIV-infected CD4^+^ T cells compared with uneducated NK cells [9,116,117]. 

HIV control in children Bw4 has been linked to reduced levels of HLA-A. In fact, levels of HLA-A in cells depends on the specific HLA-A alleles [105,107,118]. HLA-Bw4 mediates education of NK cells by its NK ligand, KIR3DL1. In fact, controller children reportedly had enriched Bw4 and co-expressed KIR3DL1, which educates these NK cells [9]. A deletion in the NKG2C gene is also associated with enhanced HIV infection, suggesting that this NK-activating receptor is a clue in the context of HIV protection [119].

Recently communicated data from several international congresses published by our group reported that LTPTC are enriched in alleles implicated in educated NK cells. Sáez-Cirión et al. found that LTPTC had certain education-related alleles such as homozygosis at HLA-Bw4 and its ligand, KIR3DL1, and also at HLAB*35, which is also an HLA-B -21 T allotype [120,121]. In fact, the HLA-B*35 allele is not a Bw4 epitope, but exceptionally is an HLA-B -21T [20,107]. These results suggest that HLA -21T (HLA B*35), Bw4 (also HLA -21T) and KIR3DL1 alleles have been associated with HIV LTPTC, suggesting an important role of NK cell education in NK-mediated HIV control [96,121]. These recent data point to a possible role of NK cells and, possibly, other innate T cells such as γδ T cells, in LTPTC HIV control process [96,122]. 

Moreover, a recent communication indicated that EEC have a high content of homozygous Bw4 alleles. In fact, 75% of EEC have two Bw4 alleles, suggesting that a high amount Bw4 homozygosis could be related to HIV functional cure [123,124,125,126,127,128,129]. Consistently, another recent communication suggests that LTEC have a high percentage of NKG2C^+^CD57^+^ NK cells [97]. Due to similar features in EEC and LTEC, such as persistent HIV control for more than 10 years, we could hypothesize that the high frequency of memory-like NK cells found in LTEC could be related to favorable NK immunogenetic characteristics. Thus, these features could be linked to HIV long-term control and functional cure. Consistent with these data and as Sáez-Cirión communicated [120], our recently published results concerning the case report of the “Barcelona patient”, reveal that this patient is a homozygote not only for the HLA haplotypes KIR3DL1/HLA-Bw4, but also for KIR2DL3/HLA-C1 (C*16) alleles. In fact, both genotypes are compatible with highly educated NK cells. Moreover, we found that this genotype is linked to an increased expansion of memory-like NK cells and γδ CD8^+^ T cell subpopulations associated with persistent HIV control and a low viral reservoir. These factors might be contributing to the observed HIV functional cure [96]. 

In summary, homozygosis at Bw4, and homozygosis at -21T and at KIR3DL1 genes is highly associated with persistent HIV control. These results may suggest that PLWH with these genetic features could educate NK cells by the KIR3DL1/HLA-Bw4 interaction having an enhanced capacity to mediate killing of HIV-infected CD4^+^ T cells (Figure 3) than the ones with uneducated NK cells. This fact suggests that not only CTL but also NK cells could have an important role in HIV control and protection. Genetic studies of NK cell education alleles related to both NKG2A/HLA-E, KIR/HLA-B interactions and HLA-A and HLA-B -21 alleles could increase the knowledge of the NK immunogenetics associated with HIV functional cure. Larger clinical trials with cohorts of LTPTC, LTEC or EEC will be necessary to explore more deeply whether protective HLA and KIR allotypes are able to potentiate NK cells against HIV infected cells by KIR3DL1/HLA-Bw4 NK education [9]. 

### 3.5. Summary of the HIV Section

In conclusion, due to the fact that dasatinib is able to induce these innate NK and γδ T cell subsets highly associated with long-term HIV control, the use of this drug could be a new strategy in the HIV functional cure field. Dasatinib could activate NK cells by up-regulation of activating receptors such as NKG2C, NKG2D, NKp46 and reducing the NKG2A/HLA-E interaction by reducing NKG2A expression in NK cells mediating HIV- infected cells killing (Figure 1B). Dasatinib could act against HIV-1 by maintaining the antiviral effect of SAMHD1 and by inhibiting CD4^+^ T-cell activation and proliferation, reducing the establishment of viral reservoirs and limiting the replenishment and maintenance of infected cells. Additionally, the potentiation of subpopulations such as memory-like NK cells and γδ CD8^+^ T cells associated with low reservoirs, persistent HIV control and having high cytotoxic potential against HIV infected cells could be key against HIV-1 infection. 

Clinical trials using dasatinib as a new immunomodulatory drug that potentiates innate cytotoxic cells could be relevant to increase the knowledge on the role of dasatinib as an NK-activating drug against HIV infection. In fact, our group has a recently approved pilot clinical trial (DASAHIVCURE NCT05527418) that will test the safety, tolerability, and antiretroviral activity of dasatinib in a cohort of recently asymptomatic HIV-infected individuals. This study will evaluate the capacity of dasatinib to inhibit HIV replication and to potentiate cytotoxic responses against HIV mediated by NK and γδ T cells.

## 4. Dasatinib as a Senolytic Drug

### 4.1. Cellular Senescence Results in Increased Immune Surveillance

Cellular senescence is activated in response to a myriad of stressors or insults. Senescent cells display a series of features that result in the recruitment and activation of immune cells with the ultimate goal of removing damaged cells and restoring tissue homeostasis. Thus, activation of cellular senescence has beneficial effects in wound healing, tissue regeneration and blocking tumor progression [130,131,132]. However, accumulation of senescent cells during aging or due to different pathologies has deleterious consequences [133].

To target senescent cells, it is key to define characteristic biomarkers, taking into consideration that these will vary depending on cell type and the type of stress [134]. The upregulation of cell cycle inhibitors such as p16^INK4a^, p21^CIP1^ and p53, or the increased expression of BCL-2, an anti-apoptotic marker, define most senescent cell types. In addition, these cells present some metabolic changes, such as an augmented activity of lysosomal senescence-associated β-galactosidase (SA-β-gal) due to a mitochondrial expansion, and also secrete specific inflammatory factors named senescence-associated secretory phenotype or SASP [133].

Immunosurveillance mechanisms are key to detect and eliminate senescent cells. A growing body of evidence points towards NK cells and their activating receptors as key cells in removing senescent cells. Gasser et al. first reported that NKG2D ligands ULBP1, ULBP2 and ULBP3 were upregulated in human fibroblasts after ionizing radiation or inhibitors of DNA replication, stimuli that activate the ATM/ATR DNA damage response and are now known to induce cellular senescence [135].

Krizhanovsky’s group [136] found that senescent cells accumulate in the liver of mice treated with CCl_4_, a chemical known to induce liver fibrosis. Livers from p53 and INK4a/ARF double knockout animals showed up to a 50% increase in liver fibrosis markers after CCL4 treatment compared to WT animals, with little to no accumulation of senescent cells. While WT animals fully resolved liver fibrosis and eliminated senescent cells in 20 days after treatment, double KO animals were severely impaired in doing so, underscoring that inducing cell senescence is important for liver fibrosis resolution [136]. Through gene profiling of senescent hepatic stellate cells (HSCs), authors show that many genes upregulated in those cells mediate an increased NK cell function, including CD58, IL-8 and NKG2D ligands such as MICA, ULBP2. Depletion of NK cells in these mice models resulted in delayed fibrosis resolution and incomplete senescent cells’ removal. Conversely, boosting NK cell function with Poly I:C (TL3-agonist) resulted in decreased fibrosis and reduced senescent cell persistence. Several studies have expanded this knowledge, showing that a wide variety of senescent cells express increased levels of NKG2D ligands, such as MICA, ULBP1 and ULBP2, in response to DNA damage (etoposide), telomere shortening (replicative renascence) or H-RAS^v12^ overexpression (oncogene-induced senescence) [137]. Further, pharmacological blockade or gene knockdown (siRNA) of NKG2D interaction with MICA or ULBP2 ligands reduces NK-mediated cytotoxicity towards senescent cells by 80–90% in vitro [137], confirming the important role of NKG2D and NK cells in removing senescent cells.

Interestingly, a recent study shows that activated invariant natural killer T cells (iNKT) are able to target and eliminate senescent cells both in a human in vitro setting and in two different mouse in vivo models [138]. Authors report that beta-2-microglobulin (B2M) and CD1d are upregulated in senescent cells, forming a complex that activates iNKT cells [138].

It is widely known that senescent cells accumulate with chronological age and that immune function also declines over time. Recent studies show that perforin release and binding (a key effector mechanism of NK and other cytotoxic cells) in the immunological synapse declines with age [139]. This is further modeled in *prf^−/−^ KO* mice, which lack perforin. These mice show an up to 2- to 4-fold increase in senescent cells across a variety of tissues, resulting in chronic inflammation and increased age-related disorders, showing that cytotoxic activity has an important role to avoid excessive senescent cell accumulation.

All of this body of evidence suggests that NK cells could be harnessed as a therapeutic option to remove senescent cells. To this end, [140] has already shown that infusion of autologous, in vitro expanded and activated NK cells (1 × 10^9^) can reduce p16^INK4a^ and SA-β-gal in PBMCs up to 90 days after infusion.

### 4.2. NK Cells Can Remove Senescent Tumor Cells

In the cancer research setting, the contribution of cellular senescence to immune surveillance is demonstrated by reactivation of endogenous p53 in p53-deficient tumors [141]. Briefly, reactivating p53 in such tumors causes cell senescence, in vitro cell cycle arrest, and triggers an NK-mediated immune response that targets tumor cells in vivo, resulting in tumor regression. Further studies using the same p53 reactivation model in mice showed that the elimination of senescent tumor cells depend on NKG2D expression. In addition, p53 is key for senescent cells to secrete pro-inflammatory chemokines such as CCL2 that recruit NK cells to senescent cells [142]. A blocking antibody against CCL2 prevented NK cell recruitment to senescent tumors, reducing their elimination. This contribution of the SASP for an optimal NK cell function was also revealed in a model of therapy-induced tumor senescence, where targeting the p65 subunit of NF-κB reduced senescent SASP secretion and proper NK-cytotoxic function [143]. 

The fact that senescent cells are prone to be detected and cleared by the immune system prompted researchers to test inducing cell senescence in tumors to address whether malignant cells can be then eliminated by the immune system. New therapeutic avenues are being studied to sensitize cancer cells to NK-mediated killing. In this regard, the chemotherapeutic drug doxorubicin, which is also a senescence-inducing drug, is able to improve NK-mediated killing of the MCF7 breast cancer cell line, through upregulation of death receptors FASR [144].

### 4.3. Senescent Cells Can Avoid NK-Cell Recognition, Thwarting Immune Clearance

While activating NK ligands are clearly upregulated in senescent cells, these cells may also upregulate inhibitory signals, resulting in the inhibition of NK-cell function. In this regard, Pereira et al. elegantly showed that primary human dermal fibroblasts were made senescent by ionizing radiation upregulating HLA-E [145]. This results in NK cell and T CD8^+^ cell inhibition as a result of NKG2A ligation in those cells. HLA-E upregulation seems greatly dependent on SASP mediators such as IL-6 and the activation of the P38 signaling pathway. Authors confirm that NKG2D is key for NK-mediated killing of senescent cells, as blocking NKG2D with a monoclonal antibody abrogates NK-mediated killing of senescent fibroblasts. Conversely, authors showed for the first time that blocking or inhibiting NKG2A with siRNA actually boosted NK-mediated senescent killing.

This underscores the key role of these receptors in senescent or tumor cell removal [146]. Additional mechanisms of NK-cell resistance include the SASP-mediated recruitment of CCR2^+^ immature myeloid cells (iMC), which inhibit NK-cell inhibition in established tumors, thus further contributing to tumor growth [147,148].

### 4.4. Dasatinib as a Senolytic Therapy in Animal Models and Human Models

The senolytic effect of dasatinib was first reported by Zhu et al., after a transcript analysis of preadipocytes made senescent by ionizing radiation (10 Gy) [149]. Further analysis showed increased negative regulators of apoptosis and anti-apoptotic gene sets in senescent cells such as ephrin receptors, the BCL-2/BCL-XL family, P13K/AKT, HIF-1a and others. Out of a panel of 46 drug candidates targeting these pathways, dasatinib and quercetin (D+Q) showed promise in selectively eliminating senescent preadipocytes, although with much reduced effect in other senescent cell types [149]. Importantly, the broad-spectrum tyrosine kinase inhibition mediated by dasatinib interferes with the ephrin receptor family (EFNB), which are the most extended tyrosine kinase receptor family and mediate key antiapoptotic signals [149].

In Zhu et al.’s work, in vivo administration of D+Q reduced the senescent cell burden in chronologically aged, radiation-exposed, and progeroid Ercc1^−^/D mice, delaying age-related symptoms and pathologies [149]. A similar effect was observed by Xu et al., when they found a healthier, longer mouse lifespan as a result of a senolytic therapy with dasatinib [140]. Since this first observation, D or D+Q senolytic strategies have been successfully used in a wide range of mouse models of diseases such as lung fibrosis [150], renal fibrosis [151], T2 diabetes [152] vascular pathology and atherosclerosis [153]. In all these models, D+Q reduced the senescent cell burden and improved organ function and physical health of treated animals. In the mouse setting, intestinal inflammation was reduced after D+Q intervention [154]. These D+Q effects have also been addressed in zebrafish models of skin inflammation and fatty liver disease, also resulting in reduced inflammation after therapy [155]. 

There are two early phase clinical trials with published results on D+Q senolytic therapy. The first in-human open-label pilot study (NCT02874989) showed that intermittent D+Q treatment (100 mg D + 1250 mg Q over three consecutive days in three consecutive weeks) is feasible, well tolerated and can alleviate physical dysfunction in idiopathic pulmonary fibrosis [156]. The second study (NCT02848131) showed that a single, 3-day dose of D+Q significantly reduced senescent cell burden (p16^INK4a^, P21^CIP^ and SA-β-gal positive cells) and plasma pro-inflammatory SASP mediators in diabetic kidney disease [157].

Different clinical trials are currently ongoing comparing the D + Q senolytic strategy to other drugs such as Fisetin in a variety of conditions, ranging from Alzheimer disease (NCT04063124) to hematopoietic stem cell transplant survivors (who are at risk of premature aging) (NCT02652052) and skeletal muscle health in older individuals (NCT04313634). More clinical trials are under way [158], which will shed light on whether senolytic D+Q treatments can improve a number of conditions.

In addition to its senolytic effects, D+Q therapy has also been shown to reduce the secretion of pro-inflammatory molecules and SASP mediators. In this regard, D+Q reduced the secretion of pro-inflammatory cytokines in human adipose tissue explants from obese individuals [153]. In the HIV context, our own work shows that D+Q senolytic therapy is able to reduce senescence biomarkers in PBMCs derived from PLWH [159]. 

While NK function is key to remove senescent cells both in vitro and in vivo, we are not aware of studies addressing whether senolytic treatment (dasatinib or others) may impact NK cell-mediated senescent cell clearance. This would be valuable to better understand the beneficial effects of dasatinib senolytic treatments. Furthermore, while promising, D+Q senolytic therapy may not be effective in all in vitro models, underscoring the importance of the experimental setting and cell type chosen when evaluating D+Q effects [149,160,161]. 

Results of ongoing studies and clinical trials will give clues to demonstrate if D+Q remains effective against aging related pathologies and could be translated towards new effective and safe anti-aging therapies.

### 4.5. Summary of the Senescence Section

Dasatinib treatment is effective in removing senescent cells in a wide variety of conditions, as shown in human clinical models. In addition, more evidence points to NK cells being key in senescent cell clearance and immunosurveillance. Dasatinib has the potential to activate NK cells by downregulating inhibitory receptors such as NKG2A and increasing activating receptors such as NKG2C and NKG2D. Thus, dasatinib may potentiate the NK cytotoxic effect against senescent cells. More research is needed in the aging field to attempt to elucidate this open question and the interesting therapeutic avenue (Figure 1C).

## 5. Concluding Remarks and Perspectives

Dasatinib treatment in CML induced populations of NK memory-like cells, γδ T cells and other innate T cells, which are associated with better prognosis of CML, a possible indefinite TFR and good cellular response. Knowledge on the precise mechanisms by which dasatinib treatment yields these immunomodulatory effects is important, as this may improve treatment interruption parameters in CML, but also help to fight against other malignancies (Figure 1A). 

A future goal is to explore whether dasatinib could safely prevent HIV-1 replication in patients both directly inhibiting SAMHD1-phosphorilation and also indirectly promoting the expansion of NK (NKG2C^+^CD57^+^) cells, γδ T cells and other innate T cells. These cell subsets show a highly effective anti-HIV response and seem to be expanded in CMV^+^ individuals. Recently, our published data concerning the case of a functionally cured HIV patient suggest that there are genetic features associated with the expansion of these innate NK and γδ T cell subsets that allow HIV persistent control. Consequently, strategies able to expand these subpopulations are important to potentiate NK cytotoxicity against HIV. This would be extremely valuable in PLWH without these genetic characteristics and may pave the way to achieve a possible functional cure. The use of dasatinib in combination with ART or new combined immunotherapies could protect CD4^+^ T cells and macrophages from HIV infection and activate a powerful cytotoxic response that could promote the elimination of the viral reservoirs (Figure 1B) and also reduce HIV-associated inflammation and senescence. 

Finally, dasatinib shows potential against inflammation and cellular senescence or aging (Figure 1C). While most senolytic effects are reported in synergy with quercetin, dasatinib on its own downregulates NKG2A and increases NKG2C and -D in NK cells. Thus, dasatinib could potentiate the already important role of NK cells in removing senescent cells, which may be an important contributor to the overall senolytic effects. More research is needed in the field to attempt to elucidate this open question.

## Figures and Tables

**Figure 1 pharmaceutics-15-00917-f001:**
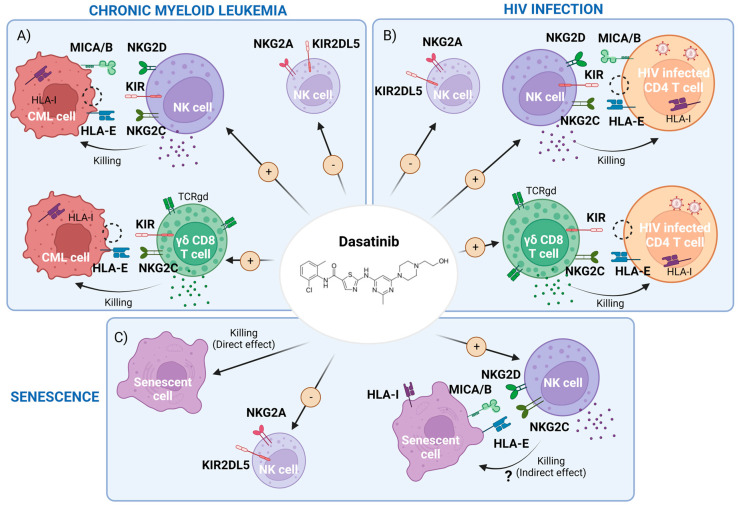
Scheme of dasatinib-mediated effects that may interfere with chronic myeloid leukemia (CML), HIV-1 infection and cellular senescence. (**A**) CML cell killing by the enhancement of memory natural killer (NK) cells and cytotoxic CD8^+^ T cells expressing γδTCR and a reduction of inhibitory receptors driven by dasatinib. (**B**) HIV infected cells’ clearance, carrying pro-viral DNA, by dasatinib activity through the potentiation of the above cell subpopulations. (**C**) Representation of both direct killing and a potential indirect effect by boosting NK and CD8^+^ cells of senescent cells with dasatinib. Figure made with BioRender.com (accessed on 7 March 2023).

**Figure 2 pharmaceutics-15-00917-f002:**
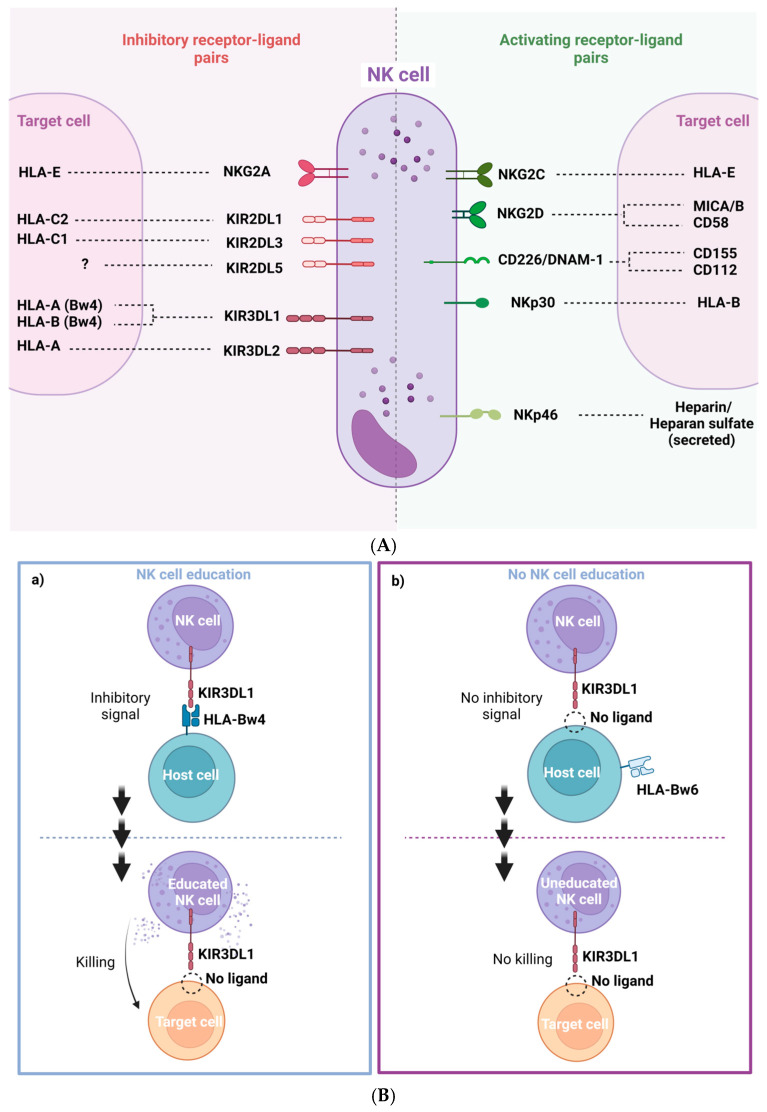
(**A**) Representation of NK cell inhibitory and activating receptor-ligand pairs. Adapted from [9]. Figure made with BioRender.com. (**B**) NK cell education process and “missing self” killing of target cells. (**a**) NK cells are recognized through KIR3DL1 receptor host cells expressing HLA-Bw4 and become educated to recognize its absence then in target cells. This process leads to an efficient target cell killing. (**b**) Absence of recognition of HLA-Bw4 by KIR3DL1 drives a deficient cell killing of target cells with “missing self” by miseducation of the NK cell. Adapted from [12]. Figure made with BioRender.com. (**C**) Genomic organization of KIR A and B haplotypes. Adapted from [12].

**Figure 3 pharmaceutics-15-00917-f003:**
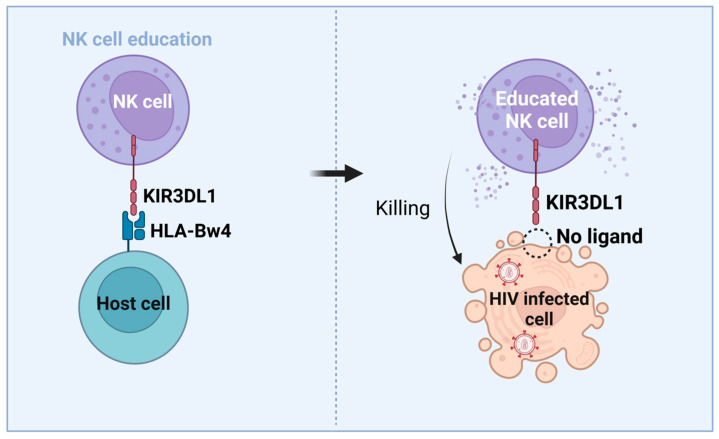
NK cell education outline in HIV context by KIR3DL1/HLA-Bw4, associated with HIV infection control. Adapted from [9]. Figure made with BioRender.com.

**Table 1 pharmaceutics-15-00917-t001:** Tyrosine kinases (TKs) targeted by dasatinib and the cellular processes in which they are involved. Adapted from Araujo et al., 2010 [5]. Abbreviations: PDGFRβ, platelet-derived growth factor receptor beta; EPHA2, ephrin type-A receptor 2; EFNB, ephrin-B; M-CSF, macrophage colony-stimulating factor.

Tyrosine Kinase Target (s)	Pathways and Processes
SRC family (SRC, LCK, YES, FYN)	Oncogenic, invasive and bone-metastatic processes
BCR-ABL	Promotion of growth advantage of leukemic cells
c-KIT	Cell growth
PDGFRβ	Tumor growth capacity and cell survival
c-FMS	Macrophage behavior regulation by M-CSF
EPHA2 receptor	Interference with EFNB-dependent suppression of apoptosis/Cell behavior

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
