# Peer review of "Immunomodulatory Activity of the Tyrosine Kinase Inhibitor Dasatinib to Elicit NK Cytotoxicity against Cancer, HIV Infection and Aging"

_pharmaceutics, 2023, doi:10.3390/pharmaceutics15030917_

Round 1
Reviewer 1 Report
1. Line 27, I personally disagree dasatinib would provide “cure” to CML, and this could be an over interpretation. My rationales are: first dasatinib has been approved since 2006, CML is not cured yet, and more TKIs are getting approved for the same disease. Second, cancer resistance mechanism such as mutation development would lead to drug resistant and disease progression.
Author Response
Answer to the referees’ and editor’s comments  
 
Comments and Suggestions for Authors  
 
Reviewer 1 
 
 
1.     Line 27, I personally disagree dasatinib would provide “cure” to CML, and this could be an over interpretation. My rationales are: first dasatinib has been approved since 2006, CML is not cured yet, and more TKIs are getting approved for the same disease. Second, cancer resistance mechanism such as mutation development would lead to drug resistant and disease progression. 
 
Answer: Thank you for your suggestion to change the word “cure” for other more suitable, such as “improving….outcome”. We have corrected these words (line 41-42 from the attached "Track changes" manuscript version). 
Thank you for your exhaustive revision that will highly improve the quality of the manuscript. 
 

Reviewer 2 Report
Rodríguez-Agustín et al. reviewed the immunomodulatory activity of the tyrosine kinase inhibitor 2 dasatinib to elicit NK cytotoxicity against cancer, HIV infection, and aging. This review article is exciting and organized well. I have some minor concerns which need to be addressed.
1. Authors should discuss the clinical outcomes of dasatinib in detail (introduction part)
2. Concluding remarks should be revised.
3. Side effects of dasatinib due to the off-target impacts should also be discussed.
4. Typo-errors should be corrected.
Author Response
Answer to the referees’ and editor’s comments  
 
Comments and Suggestions for Authors 
Reviewer 2# 
 
Rodríguez-Agustín et al. reviewed the immunomodulatory activity of the tyrosine kinase inhibitor 2 dasatinib to elicit NK cytotoxicity against cancer, HIV infection, and aging. This review article is exciting and organized well. I have some minor concerns which need to be addressed. 
 
1.      Authors should discuss the clinical outcomes of dasatinib in detail (introduction part). 
 
Answer:  We would like to thank the reviewer for the positive comments on the manuscript and the suggestions made. We have now detailed the clinical outcomes of dasatinib in the introduction section, lines 56-64 from the attached "Track changes" manuscript version.  
 
2.      Concluding remarks should be revised. 
 
Answer: Thank you for your indication. We have revised this section (lines 616-648).  
 
3.      Side effects of dasatinib due to the off-target impacts should also be discussed. 
 
Answer: We have now included information regarding off-target effects in the dasatinib introduction section in lines 67-73. In addition, we have included two related references in this section (Araujo 2010 492) (Cheng F, et al. 2023 1113462). 
 
4.      Typo-errors should be corrected.  
 
Answer: We have corrected the typo-errors from all the manuscript.  
Thank you for your exhaustive revision that will highly improve the quality of the manuscript. 
 

Reviewer 3 Report
The manuscript "Immunomodulatory activity of the tyrosine kinase inhibitor dasatinib to elicit NK cytotoxicity against cancer, HIV infection and aging", written by Rodriguez-Agustin A, Casanova V, Grau-Exposito J, Sanchez-Palomino S, Alcami J. and Climent N. presents the recent results of the experiments showing the effects of dasatinib on the activity o NK cells and consequences of these processes. Dasatinib is a broad-spectrum tyrosine kinase inhibitor, used for treatment of chronic myeloid leukemia (CML) and found to have numerous additional effects. In this manuscript, the accent is on three fields: dasatinib effects on NK cells in CML treatment, potential in HIV infection therapy and role in decreasing the number of senescent cells, therefore influencing the aging.
The manuscript contains lot of up to date information and experiment details tightly linked to the topic. The Introduction presents the main roles of dasatinib, also illustrated with the figure. However, I miss the list of kinases or signaling pathways and processes which it inhibits. Second part presents the biology of natural killer cells, which are the topic of the manuscript. It is very detailed, although part presenting "KIR haplotypes" could be described more clearly. Also, a mechanism of NK education and deep molecular response could be explained. Further on, interplay among dasatinib, CML and NK cells is described and summarized at the end. Second part describes the potential role of dasatinib in HIV infection therapy, from drug effects on the cell enzymes to effects on NK response. Possibly the summary could be written in the form of the text. The third part is considering dasatinib as a senolytic. This part also has several paragraphs, from those explaining the process of senescence to experiments involving senolytics and showing that NK cells could have a role in that processes. The last part describes dasatinib as a senolytic drug, but there are no citations of roles of NK cells and molecular mechanisms in the processes of ameliorating different aging diseases in animal models (paragraph 4). It seems to me that this part is off-topic. In Concluding remarks, repetition of the same sentences as in the main text should be avoided.
Other comments:
line 24: explain expansion of cells
Dasatinib is written with small and capital letters, it should be standardized
line 53: if NK cells have to be explained as natural killer cells, it should be done first in the Abstract and in Fig. 1
Figure 2A: what is the right name of the author from reference
Heparin and heparan sulphate are extracellular molecules
on several places references were left in another format (author + year)
sentence reorganization or tipfellers: lines 23, 58, 74, 115, 139, 141, 159, 175, 179, 182, 183, 198, 225, 229 244, 270, 273, 275, 294, 321, 325, 329, 342, 351, 377, 413, 417, 429, 433, 437, 457, 457, 458, 505
line 76: ratio instead of equilibrium?
line 69: explanation of reprogrammed cells, degranulation,
line 89: better explanation of "enhance HLA-E on the cell membrane"
line 88: explanation of HLA leader sequences
line 120: number or percentage, instead of frequency?
line 225: explanation of CAR-T therapy
line 256: it is said that dasatinib preserved SAMHD1 activity; it could be understood that it does not act on it, but it seems to me that it keeps it active
line 266: explanation of increasing of HIV2 LTR cycles
line 275: explanation of the abbreviation PLWH
line 313: reference for "LTPTC have a NK mediated HIV control"
Author Response
Answer to the referees’ and editor’s comments  
 
Comments and Suggestions for Authors  
Reviewer 3# 
 
The manuscript "Immunomodulatory activity of the tyrosine kinase inhibitor dasatinib to elicit NK cytotoxicity against cancer, HIV infection and aging", written by Rodriguez-Agustin A, Casanova V, Grau-Exposito J, Sanchez-Palomino S, Alcami J. and Climent N. presents the recent results of the experiments showing the effects of dasatinib on the activity on NK cells and consequences of these processes. Dasatinib is a broad-spectrum tyrosine kinase inhibitor, used for treatment of chronic myeloid leukemia (CML) and found to have numerous additional effects. In this manuscript, the accent is on three fields: dasatinib effects on NK cells in CML treatment, potential in HIV infection therapy and role in decreasing the number of senescent cells, therefore influencing the aging. 
  
The manuscript contains lot of up to date information and experiment details tightly linked to the topic. The Introduction presents the main roles of dasatinib, also illustrated with the figure.  
 
- However, I miss the list of kinases or signaling pathways and processes which it inhibits.  
 
Answer: We would like to thank the reviewer for thoroughly revising the manuscript. We appreciate the detailed list of changes and suggestions made, as they will significantly improve the final manuscript. We have now introduced a Table 1, which lists the main tyrosine kinases that are inhibited by dasatinib together with its corresponding reference in the introduction section (lines 68 and 1211 from the attached "Track changes" manuscript version).  
 
- Second part presents the biology of natural killer cells, which are the topic of the manuscript. It is very detailed, although part presenting "KIR haplotypes" could be described more clearly. 
 
Answer: Thank you for your advice on clarifying KIR haplotypes’ section. We revised both the NK introduction section and the Figure 2C, adding a sentence which helps at describing more clearly the KIR gene distribution (lines 103-111 1247). 
 
  
- Also, a mechanism of NK education and deep molecular response could be explained.  
 
Answer: Thank you for your suggestion. We have included information explaining the potential mechanism of NK education and deep molecular response in CML (lines 249-264). 
 
 
Further on, interplay among dasatinib, CML and NK cells is described and summarized at the end.  
 
- Second part describes the potential role of dasatinib in HIV infection therapy, from drug effects on the cell enzymes to effects on NK response. Possibly the summary could be written in the form of the text.  
 
Answer: Thank you for your recommendation. We have changed the summary as suggested (lines 439-462).  
 
- The third part is considering dasatinib as a senolytic. This part also has several paragraphs, from those explaining the process of senescence to experiments involving senolytics and showing that NK cells could have a role in that processes. The last part describes dasatinib as a senolytic drug, but there are no citations of roles of NK cells and molecular mechanisms in the processes of ameliorating different aging diseases in animal models (paragraph 4). It seems to me that this part is off-topic.  
 
Answer:  Thank you for your observation. We state in the previous section that NK cells are key to remove senescent cells, but to our knowledge, no studies have addressed or reported whether senolytic effects by dasatinib are mediated by NK cell actions. We still think this section is informative at describing the current clinical trials in which dasatinib is being used (lines 593-602). 
 
 
- In Concluding remarks, repetition of the same sentences as in the main text should be avoided. 
 
Answer: We have modified these duplicate sentences as suggested (lines 616-646). 
 
 
Other comments: 
 
- line 24: explain expansion of cells.  
 
Answer: We have changed the word “expanded” into the word increased (line 37). 
 
- Dasatinib is written with small and capital letters, it should be standardized.  
 
Answer: Thank you, it has been unified all long the document. 
 
- line 53: if NK cells have to be explained as natural killer cells, it should be done first in the Abstract and in Fig. 1.  
 
Answer: Thank you, it is done. 
 
- Figure 2A: what is the right name of the author from reference.  
 
Answer: Thank you, it is amended. 
 
- Heparin and heparan sulphate are extracellular molecules.  
 
Answer: Thank you, the Figure 2A has been modified. 
 
- on several places references were left in another format (author + year).  
 
Answer: Thank you, it is done. However, as the editor suggested, we would change the references format at the last without “track-changes” version of the manuscript and after your second revision.  
 
- sentence reorganization or tipfellers: lines 23, 58, 74, 115, 139, 141, 159, 175, 179, 182, 183, 198, 225, 229 244, 270, 273, 275, 294, 321, 325, 329, 342, 351, 377, 413, 417, 429, 433, 437, 457, 457, 458, 505 
 
Answer: We have had a problem localizing all these tipfellers because the lines from the version of the manuscript found at the provided link, do not correspond to the lines suggested. We have done our best in order to review and correct all the tipfellers all along the manuscript.   
 
- line 76: ratio instead of equilibrium?  
 
Answer: Thank you, it is amended (line 97) 
 
- line 69: explanation of reprogrammed cells, degranulation.  
 
Answer: Thank you, it is corrected (line 234-237). 
 
- line 89: better explanation of "enhance HLA-E on the cell membrane".  
 
Answer: Thank you, it is corrected (line 113-116). 
 
- line 88: explanation of HLA leader sequences.  
 
Answer: Thank you, it is corrected (lines 113-116). 
 
- line 120: number or percentage, instead of frequency?.  
 
Answer: Thank you, it is corrected (line 146-147). 
 
- line 225: explanation of CAR-T therapy.  
 
Answer: Thank you, it is explained (lines 273-279). 
 
- line 256: it is said that dasatinib preserved SAMHD1 activity; it could be understood that it does not act on it, but it seems to me that it keeps it active.  
 
Answer: Thank you, it is better explained (line 306-312). 
 
- 15. line 266: explanation of increasing of HIV2 LTR cycles.  
 
Answer: Thank you, it is corrected (line 323-325). 
 
- line 275: explanation of the abbreviation PLWH.  
 
Answer: Thank you, it is corrected (lines 334-335). 
 
- line 313: reference for "LTPTC have a NK mediated HIV control"  
 
Answer: Thank you, it is added (lines 410). 
